# Herbal formula PM012 induces neuroprotection in stroke brain

**Kuo-Jen Wu[1], Yu-Syuan Wang[1], Tsai-Wei Hung[1], Eun-Kyung Bae[1], Yun-Hsiang Chen[2], Chan-Kyu Kim[3], Dai-Won Yoo[3], Gyeong-Soon Kim[3], Seong-Jin Yu[1]***

1 Center for Neuropsychiatric Research, National Health Research Institutes, Zhunan, Taiwan,
2 Department of Life Science, Fu-Jen Catholic University, New Taipei City, Taiwan, 3 Mediforum Co., Ltd., Seoul, Republic of Korea

* b7508@nhri.edu.tw

## Abstract

Stroke is a major cause of long-term disability world-wide. Limited pharmacological therapy has been used in stroke patients. Previous studies indicated that herb formula PM012 is neuroprotective against neurotoxin trimethyltin in rat brain, and improved learning and memory in animal models of Alzheimer's disease. Its action in stroke has not been reported. This study aims to determine PM012-mediated neural protection in cellular and animal models of stroke. Glutamate-mediated neuronal loss and apoptosis were examined in rat primary cortical neuronal cultures. Cultured cells were overexpressed with a Ca++ probe (gCaMP5) by AAV1 and were used to examine Ca++ influx (Ca++i). Adult rats received PM012 before transient middle cerebral artery occlusion (MCAo). Brain tissues were collected for infarction and qRTPCR analysis. In rat primary cortical neuronal cultures, PM012 significantly antagonized glutamate-mediated TUNEL and neuronal loss, as well as NMDA-mediated Ca++i. PM012 significantly reduced brain infarction and improved locomotor activity in stroke rats. PM012 attenuated the expression of IBA1, IL6, and CD86, while upregulated CD206 in the infarcted cortex. ATF6, Bip, CHOP, IRE1, and PERK were significantly down-regulated by PM012. Using HPLC, two potential bioactive molecules, paeoniflorin and 5-hydroxymethyl-furfural, were identified in the PM012 extract. Taken together, our data suggest that PM012 is neuroprotective against stroke. The mechanisms of action involve inhibition of Ca++i, inflammation, and apoptosis.

## Introduction

PM012, also known as Gugijihwang-Tang, is a herbal formula with neuroactive effects [1, 2]. PM012 prevented neurotoxin trimethyltin -mediated reduction of glucose metabolism in the whole brain or hippocampus, and improved learning and memory in adult rats [1]. In an animal model of Alzheimer's disease, PM012 increased BDNF expression, enhanced BrdU and DCX labeling in hippocampus, and ameliorated memory deficit in 3XTg mice [2]. Oral administration of PM012 reduced escape latency in human presenilin 2 mutant transgenic mice [3]. These data suggest that PM012 is neuroprotective and neuroreparative in the CNS. Its mechanism of action is not clear.

**Data Availability Statement:** This study was supported in part by Mediforum Co., Ltd., Seoul, Republic of Korea (C10-020), Ministry of Science and Technology, Taiwan (MOST 110-2320-B-400-

007), Ministry of Health and Welfare, Taiwan (CS-111-GP-02).

**Funding:** This study was supported in part by Mediforum Co., Ltd., Seoul, Republic of Korea (C10-020), Ministry of Science and Technology, Taiwan (MOST 110-2320-B-400-007), Ministry of Health and Welfare, Taiwan (CS-111-GP-02).

**Competing interests:** C.K.K., D.W.Y., and K.S.K. are employees of Mediforum Co., Ltd. The National Health Research Institute and Mediforum Co., Ltd. have a Cooperative Research and Development Agreement to develop PM012 as a treatment strategy for neurodegenerative disorders. The authors declare that this study received partial funding from Mediforum Co., Ltd., Korea. The authors further declare that Mediforum Co., Ltd. provided input into dose selection and study design as well as editorial input into the article. The funder was not involved in the writing of this article, or the decision to submit it for publication. This does not alter our adherence to PLOS ONE policies on sharing data and materials.

PM012 is composed of Corni fructus (13%), Lucii fructuw (26.5%), Rehmannia radix (26.5%), Hoelen (7%), Discoreae radix (13%), Mountain cortex radices (7%), and Alismatis radix (7%) [1, 2]. These ingredients were found to regulate degenerative reactions. Rehmanniae Radix is an antioxidant and can reduce glutamate toxicity [4]. Corni fructus [5], Discoreae radix [6], Mountain cortex radicis [7], and Alismatis radix [8] are anti-inflammatory.

Ischemic stroke accounts for 87% of stroke cases [9] and is a leading cause of adult disability world-wide [10]. Ischemic insult triggers a series of progressive neurodegeneration processes, including glutamate overflow, apoptosis, inflammation, and ER stress, which lead to cell death. Suppressing these processes prevents stroke-mediated degeneration. For example, we previously reported that 2-fucosyllactose reduced ischemia-mediated ER stress and inflammation in brain and improved locomotor behavior in stroke rats [11]. Interestingly, several active ingredients in PM012 also process anti-inflammatory or anti-ER stress properties [12, 13]. PM012 may reduce ischemic brain damage through these actions.

The purpose of this study is to examine the protective actions of PM012 in cellular and rat models of stroke. Here we reported that PM012 reduced glutamate-mediated neuronal cell loss and apoptosis in primary neuronal culture. In addition, PM012 improved behavioral function and reduced brain infarction, inflammation, and ER stress in stroke rats. Our data support the notion that PM012 protects against ischemic stroke-mediated neurodegeneration.

## Results

### Major components in PM012-determined by HPLC

PM012 (650 mg) was dissolved in methanol (5ml) and filtered. The contents in the solution were examined by HPLC. As seen in the chromatograms (Fig 1), two peaks were found at 254 nm and 280 nm, corresponding to paeoniflorin and 5-Hydroxymethylfurfural (5-HMF).

### PM012 reduced glutamate-mediated neurodegeneration in primary cortical neuronal culture

The protective effects of PM012 were first examined in rat primary cortical neurons. Treatment with glutamate (100 μM) for 48h significantly reduced MAP2 immunoreactivity (MAP2-ir; Fig 2A, 2B and 2E, p<0.001) and increased TUNEL (Fig 2F, 2G and 2J, p<0.001). PM012 (0.1mg/ml or 1mg/ml, n = 11 per each group) significantly antagonized glutamate-mediated changes in MAP2-ir (0.1mg: Fig 2C; 1mg: Fig 2D and 2E, p<0.001, one-Way

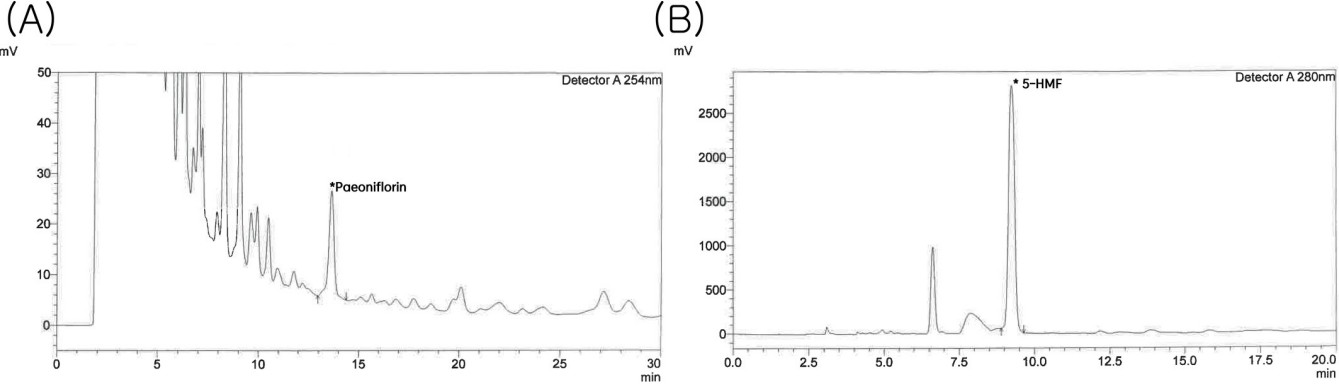

**Fig 1. Quantification of chemicals in PM012 using HPLC.** HPLC chromatograms of two standards (A) paeoniflorin and (B) 5-HMF.

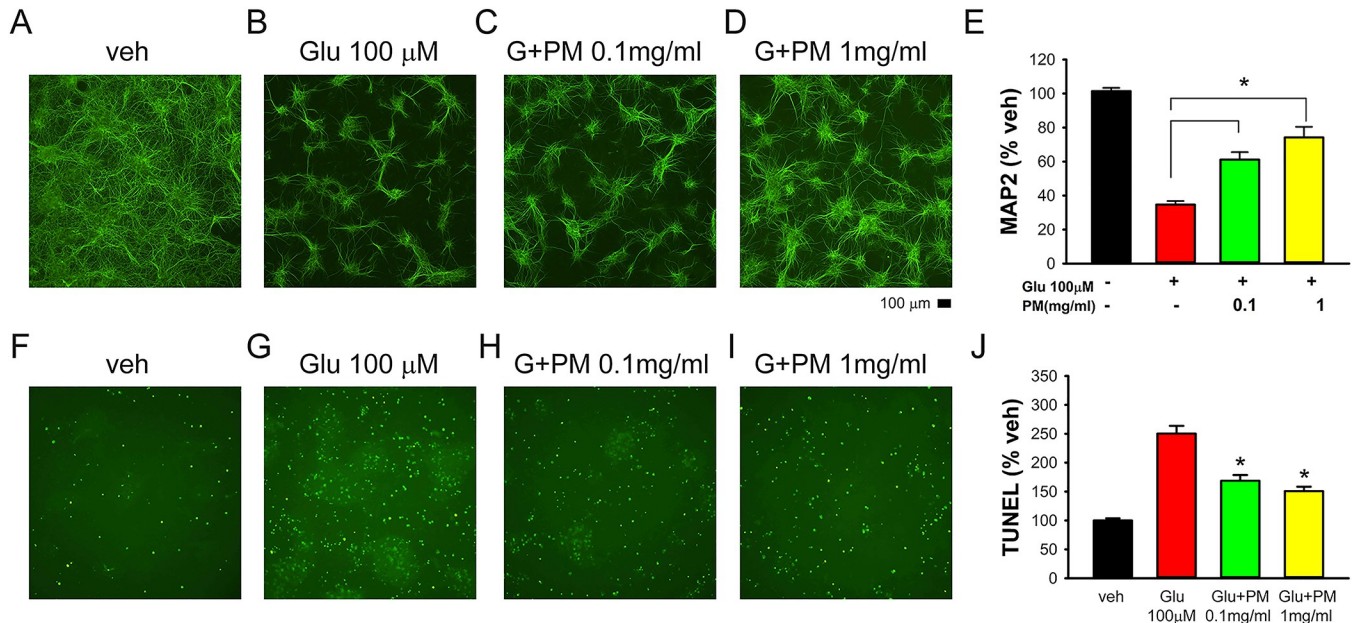

**Fig 2. The neuroprotective effect of PM012 in primary cortical neuronal culture.** Representing photomicrographs demonstrate that glutamate (Glu)-reduced (B) MAP2 immunoreactivity and induced (G) TUNEL. Co-administration with PM012 (C, 0.1mg/ml; D, 1mg/ml) attenuated Glu-mediated loss of MAP-ir. (E) PM012 significantly antagonized Glu-mediated loss of MAP2-ir. (G vs. F) Glu increased TUNEL labeling. Co-administration with PM012 (H, 0.1mg/ml; I, 1mg/ml) reduced TUNEL activity. (J) PM012 significantly attenuated Glu-mediated TUNEL activity. *p<0.05, one-Way ANOVA.

ANOVA+NK test) and TUNEL (0.1mg: Fig 2H; 1mg: Fig 2I and 2J, p<0.001, one-Way ANOVA+NK test).

## PM012 inhibited NMDA-mediated Ca$^{++}$i in primary cortical neurons

Primary cortical neurons were treated with AAV- gCaMP5 on DIV5 to express a calcium probe gCaMP5, as we previously described [14]. N-methyl-d-aspartate (NMDA, 5μM) was added to the wells on DIV12 to induce calcium ion influx (Ca++i). Real-time fluorescence images were taken 6 seconds before to 12 seconds after drug administration (Fig 3). NMDA triggered a rapid and time-dependent increase in Ca++i (Fig 3A vs. A1). PM012 suppressed NMDA -activated Ca++i (Fig 3B vs. B1). Time-dependent Ca++i was next analyzed in 44 cells (Fig 3C). PM012 significantly suppressed NMDA -mediated Ca++i (p<0.05, two-Way ANOVA+NK test).

## Pretreatment with PM012 improved locomotor activity in stroke rats

Thirteen rats received PM012 (50 mg/kg/d, x 2d, i.p.), and 9 received vehicle, starting from 2 days before the MCAo. Another 5 rats did not receive MCAo and were used as naive controls. Locomotor activity was examined 2 days after MCAo. Stroke animals developed bradykinesia. Horizontal activity (HACTV), total distance traveled (TOTDIST), and vertical activity (VACTV) were significantly reduced in stroke animals receiving vehicle (Fig 4, p<0.05, Stroke vs. naive); PM012 significantly antagonized these responses (stroke+PM012 vs. stroke+veh). Treatment with PM012 significantly increased HACTV (Fig 4A, p = 0.002), TOTDIST (Fig 4B, p = 0.010), and VACTV (Fig 4C, p = 0.033) in the stroke rats.

## Pretreatment with PM012 reduced brain infarct in stroke rats

A total of 14 stroke rats (vehicle, n = 7, PM012, n = 7) were used for brain infarction analysis. Brain tissues were collected and sectioned into 2 mm slices 2 days after MCAo. Brain

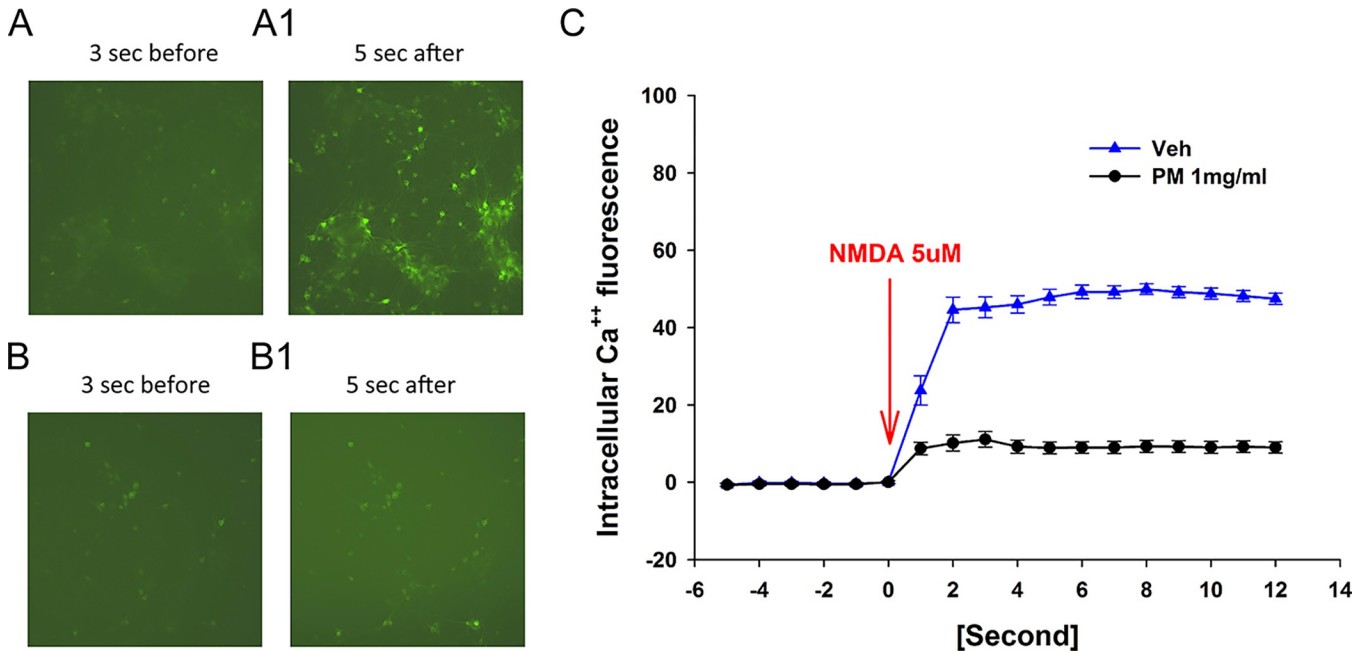

**Fig 3. PM012 suppressed NMDA -mediated intracellular Ca$^{++}$ influx in primary cortical neurons expressing GCaMP5.** Real-time Ca$^{++}$i images were taken before (A, B) and after (A1, B1) NMDA administration. NMDA (5 μM) triggered a rapid increase in intracellular Ca$^{++}$ (A1 vs. A). Co-treatment with PM012 suppressed NMDA-mediated intracellular Ca$^{++}$ signals (B1 vs. A1). (C) Intracellular fluorescence from 44 cells were analyzed. PM012 significantly suppressed NMDA -mediated Ca$^{++}$i (p<0.05).

infarction was visualized after TTC staining. Typical infarction in stroke animals receiving vehicle or PM012 is shown in Fig 5. The volume of infarction per animal was further analyzed in all animals studied. PM012 significantly reduced brain infarction (Fig 5C; p = 0.007, t-test).

## PM012 altered the expression of inflammation-related genes in stroke brains

Cortical tissues from the ischemic side (infarcted) and contralateral side (non- infarcted) hemispheres were collected from 14 stroke rats (veh, n = 7; PM012, n = 7) on day 2. The

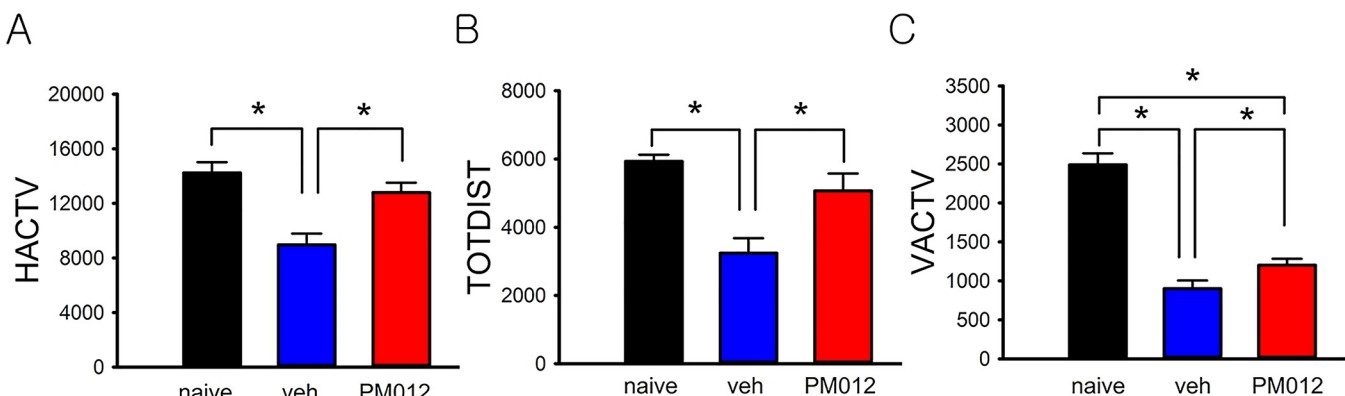

**Fig 4. PM012 improved locomotor behavioral function in ischemic rats.** Rats received intraperitoneal injections of PM012 (50mg/kg/d) or vehicle from 2 days before MCAo. Locomotor behavior was examined on day 2 after MCAo. Stroke rats (n = 9) showed a significant reduction in locomotor activity compared to naïve rats (n = 5). PM012 (n = 13) significantly improved horizontal activity (HACTV, p = 0.002), total distance traveled (TOTDIST, p = 0.010), and vertical activity (VACTV, p = 0.033) in stroke rats. *P<0.05, one-Way ANOVA+NK test.

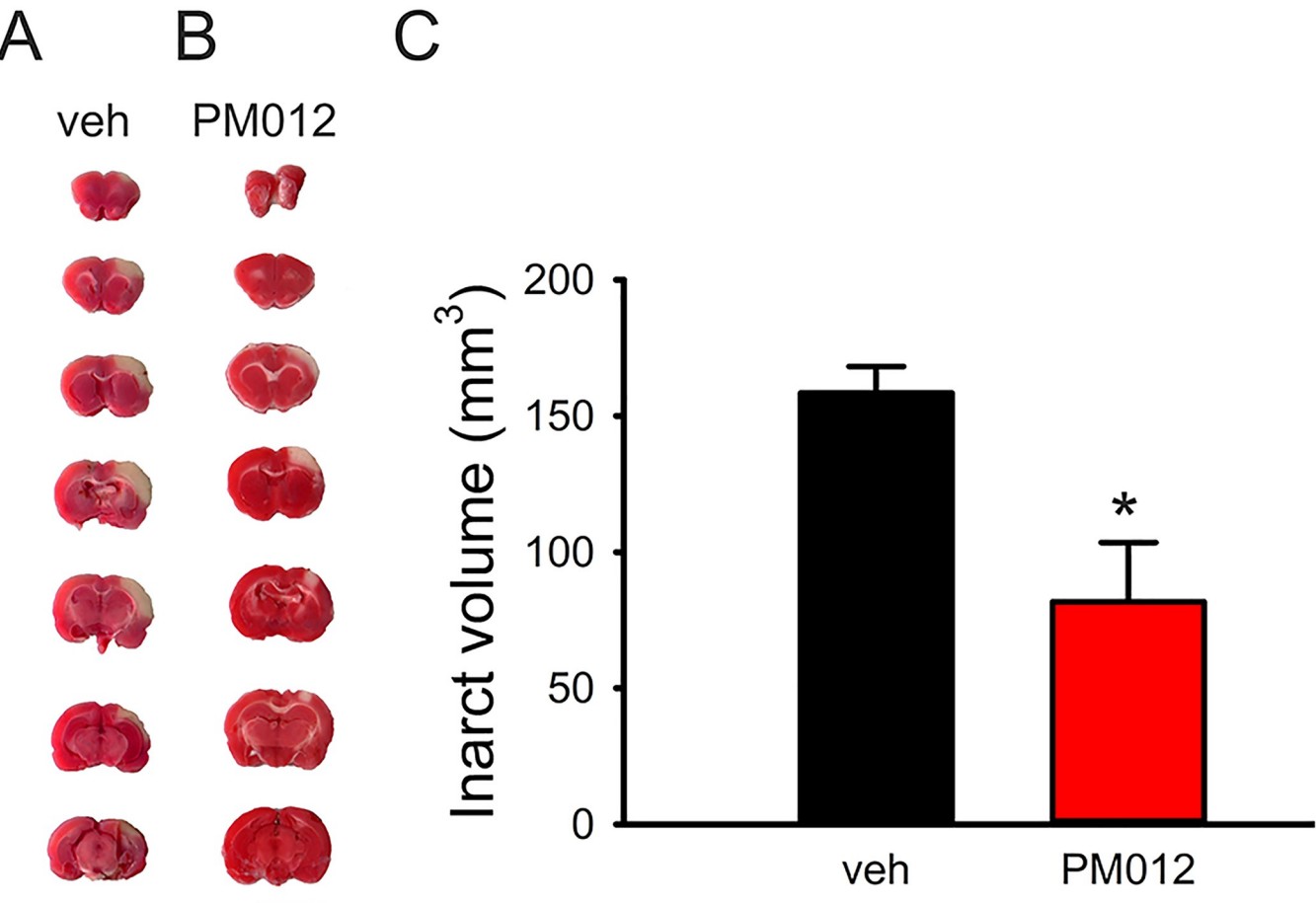

**Fig 5. PM012 reduced infraction in stroke rats.** PM012 (50mg/kg/d) was administered to animal 2 days before MCAo. Animals were sacrificed 2 days after stroke. Brain infarction was examined by TTC staining (representing animals receiving A: vehicle or B: PM012). (C) PM012 significantly reduced infarction volume (p = 0.007, t-test). Scale: 10 mm.

expression of inflammatory markers was examined by qRTPCR. Stroke increased the expression of inflammatory genes, including pro-inflammatory IBA1, IL-6, CD86, and anti-inflammatory CD206 (Fig 6, infarcted vs. non- infarcted, one-way ANOVA). PM012 significantly downregulated the expression of IBA1 (Fig 6A, p = 0.005), IL-6 (Fig 6B, p = 0.002), and CD86 (Fig 6C, p = 0.005) and upregulated the expression of CD206 (Fig 6D, p<0.001) in the infarcted cortex.

## PM012 suppressed the expression of ER stress and apoptotic genes in stroke brains

ER stress and apoptotic markers were examined by qRTPCR. The expression of ER stress (ATF6, BIP, CHOP, IRE1, PERK) and apoptotic (Caspase 3) genes were all significantly upregulated in ischemic side cortices (Fig 7, p<0.05, 1-way ANOVA). PM012 significantly reduced the expression of ATF6 (Fig 7A, p = 0.034), BIP (Fig 7B, p = 0.002), CHOP (Fig 7C, p = 0.001), IRE1 (Fig 7D, p = 0.006), and PERK (Fig 7E, p = 0.013) in the infarcted cortex. Caspase3 was significantly reduced by PM012 (Fig 7F, p<0.001). PM012 did not alter the expression of these genes in the non- infarcted side cortex.

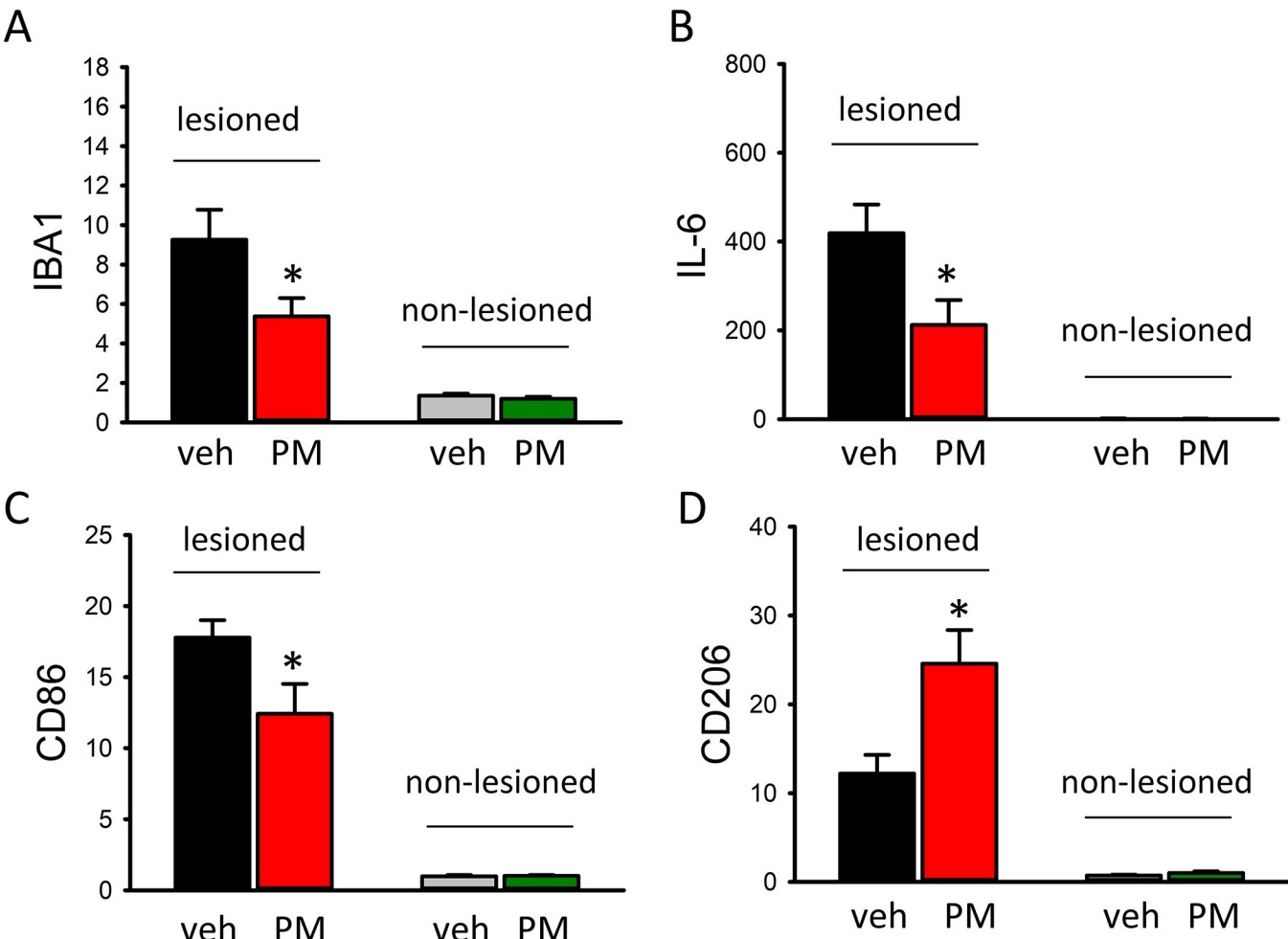

**Fig 6. PM012 altered the expression of inflammatory markers in stroke brains.** Ischemic stroke significantly increased the expression of (A) IBA1, (B) IL-6, (C) CD86 and (D) CD206 (infarcted vs. non- infarcted). PM012 significantly downregulated (A) IBA1, (B) IL-6 and (C) CD86 while upregulated (D) CD206 (Fig 6D, $p<0.001$) in the infarcted cortex. *$P<0.05$, one-Way ANOVA+NK test.

## Discussions

In this study, we demonstrated that PM012 reduced glutamate-mediated neurotoxicity, apoptosis, and Ca$^{++}$ influx in primary neuronal culture in vitro. In addition, pretreatment with PM012 reduced brain infarction, motor deficits, inflammation, and ER stress in stroke brain. The major finding of this study is that PM012 is neuroprotective against stroke.

After the ischemic injury, oxygen-rich blood supplied to the brain is reduced or blocked. Consequently, the reduction of ATP level triggers the influx of calcium ions into the pre-synaptic neurons, and results in glutamate overflow in the synapse and neurotoxicity. Using the endogenous Ca++ probe GCaMP5, we demonstrated that PM012 antagonized NMDA-mediated Ca++i in neurons. Moreover, PM012 reduced glutamate-mediated TUNEL and the loss of MAP2-ir in the neuronal culture, suggesting that PM012 induces neuroprotection through modulation of Ca++i in primary neurons.

We further examined the protective effect of PM012 in vivo. We demonstrated that pretreatment of PM012 for 2 days enhanced locomotor movements and reduced brain infarction in ischemic stroke rats, suggesting that PM012 is also neuroprotective against stroke in vivo.

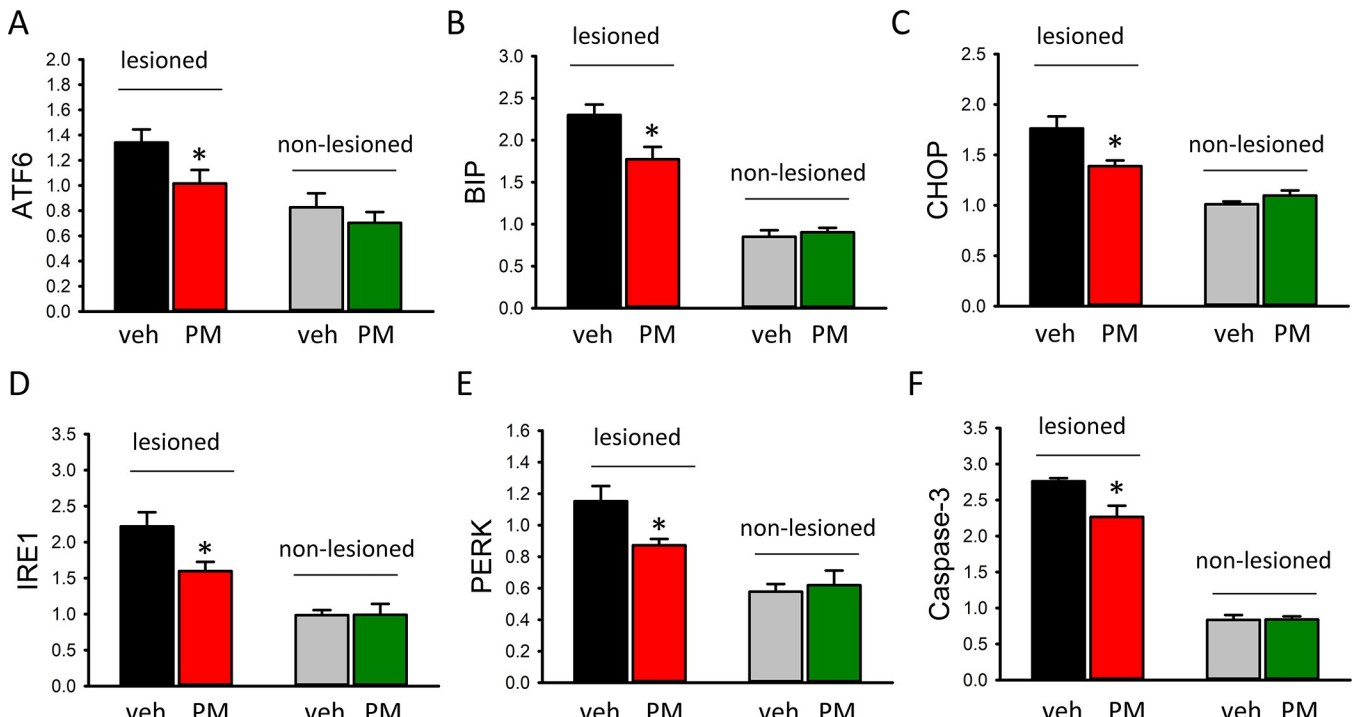

**Fig 7. PM012 suppressed ER stress and apoptosis in stroke brain.** The expression of (A) ATF6, (B) BIP, (C) CHOP, (D) IRE1, (E) PERK, and (F) Caspase3 was significantly inhibited by PM012 in ischemic stroke brains. *P<0.05, one-Way ANOVA+NK test.

Using HPLC, we identified two molecules, paeoniflorin and 5-HMF, in the PM012 extract. Previous studies have shown that 5-HMF antagonized d-galactosamine and TNF-alpha -mediated ER stress and apoptosis in cultured human hepatocytes [13]. Paeoniflorin prevented lipo-polysaccharide-induced overproduction of inflammatory marker IL6 and ER stress markers CHOP and GRP78 in human umbilical vein endothelial cells [12] and attenuated all-trans-retinal–induced ER stress in retinal pigment epithelial cells [15].

Inflammation and ER stress are major causes of neurodegeneration after ischemic brain injury. We and others reported that suppression of inflammation or ER stress improved neural functions in stroke animals [11, 16]. These data suggest that paeoniflorin and 5-hydroxy-methylfurfural are the active ingredients in PM012, which may induce protection against ischemic brain injury through anti-inflammation and anti-ER stress.

To identify the anti-inflammatory role of PM012 in stroke, we examined the expression of microglia markers in brain. Microglia play an important role in controlling the immune and inflammatory responses after ischemic brain injuries [17]. Two phenotypes (M1 and M2) of microglia with differential actions have been reported [18]. The activation of M1 microglia leads to cell death, while stimulation of M2 microglia increases cell survival [19–21]. We demonstrated that PM012 reduced the expression of IBA1, down-regulated M1 markers CD86 and IL6 [22, 23], and upregulated M2 marker CD206 [24, 25]. Our data suggest that PM012 reduced ischemic brain degeneration by modulating M1/M2 microglia in the infarcted brain.

Similar to previous studies, we demonstrated that ATF6, Bip, CHOP, IRE1, PERK, and Caspase-3 were upregulated after ischemic brain injury. PM012 significantly reduced these ER stress responses in stroke brain. Our findings were further supported by the anti-ER stress effects of two major ingredients (i.e., 5-HMF and Paeoniflorin, see Fig 1) in PM012 [12, 13,

15]. Altogether, our data suggest that PM012 reduced ischemic brain injury through the suppression of ER stress.

In conclusions, PM012 reduced neuronal degeneration, activation of microglia, ER stress, apoptosis, and cerebral infarction in stroke brain. We demonstrated that pretreatment with PM012 reduced ischemic brain damage in rats. For clinical implication, this approach (pretreatment with PM012) may be beneficial to patients with a high risk of stroke (i.e., transient ischemic attack) to prevent recurrent ischemic injury. In our future experiments, we will examine the protective effect of PM012 when given early after stroke.

## Materials and methods

### Materials

PM012 was given by the Mediforum Corporation (Seoul, Republic of Korea). Quality control for PM012 adhered to the specifications and test procedures for drugs was approved by Ministry of Food and Drug Safety in Korea.

Bovine serum albumin, sodium pentobarbital, fetal bovine serum, L-glutamate, NMDA, paraformaldehyde, polyethyleneimine, Triton X-100, and 2,3,5-triphenyl tetrazolium chloride (TTC) were purchased from the Sigma (St. Louis, USA). Alexa Fluor 488 (secondary antibody), B27 supplement, Dulbecco's modified Eagle's medium, Neurobasal Medium, and trypsin were purchased from Invitrogen (Carlsbad, USA). MAP2 antibody was purchased from the Millipore (Burlington, USA). In Situ Cell Death Detection Kit was purchased from Roche (Indianapolis, USA).

Adult male and time-pregnant Sprague-Dawley rats were purchased from BioLASCO, Taiwan. The use of animals was approved by the Animal Research Committee of the National Health Research Institutes of Taiwan (NHRI-IACUC- 109097-M1). All animal experiments were carried out in accordance with the National Institutes of Health Guide for the Care and Use of Laboratory Animals (NIH Publications No. 8023, revised 1978).

### High-Performance Liquid Chromatography (HPLC) analysis

Chromatographic analysis was performed using a LC-4000 HPLC system (JASCO, Japan) with XTerra TM $RP_{18}$ (4.6 x 150 mm, 5 μL) column (Waters, USA). The mobile phase using gradient elution consist of two solvent systems, acetic acid in water (A) and acetic acid in acetonitrile (B). The flow-rate was 1.0 ml/min and injection volume was 10 μg. The column temperature was set at 25°C.

5-HMF (0.1mg/ml) and paeoniflorin (0.1mg/ml) were dissolved in methanol and used as standards for the qualitative and quantitative analysis of PM012 (2mg/ml). Sample injection volume was 10 μL. Absorbance of column eluate was monitored with a UV spectrometer for 5-HMF at a wavelength of 280 nm and for paeoniflorin at a wavelength of 254 nm.

### Primary rat Cortical Neuron (PCN)

Primary cultures were prepared from embryonic (E14–15) cortex tissues obtained from fetuses of timed pregnant rats. The olfactory bulbs, striatum, and hippocampus were removed aseptically, and cortices were dissected. After removing the blood vessels and meninges, pooled cortices were trypsinized (0.05%) for 20 min at room temperature. After rinsing off trypsin with prewarmed Dulbecco's modified Eagle's medium, cells were dissociated by trituration, counted, and plated into 96-well ($5.0 \times 10^4$/ well) cell culture plates pre-coated with polyethyleneimine. The culture plating medium consisted of Neurobasal Medium supplemented with 2% heat-inactivated fetal bovine serum (FBS), 0.5 mM L-glutamine, 0.025-mM L-glutamate,

and 2% B27. Cultures were maintained at 37˚C in a humidified atmosphere of 5% $CO_2$ and 95% air. The cultures were fed by exchanging 50% of media with feed media (Neurobasal Medium) with 0.5 mM L-glutamate and 2% B27 with antioxidants supplement on days in vitro (DIV) 3 and 5. PM012 (20 mg) was dissolved in saline (1 mL) and filtered. The solution was further diluted in culture media to various concentrations and added to the culture wells.

## Real-time intracellular Ca++ measurement in primary neuronal culture

Primary cortical neuronal cultures were infected by AAV-GCaMP5 ($3 \times 10^{11}$ viral genome/ml) on DIV 5 for 1 h. NMDA-mediated $Ca^{++}$ influx was examined on DIV12 as previously described [14]. In brief, culture plates were placed on a motorized stage (Prior Scientific Inc., Fulbourn, Cambridge, UK) of a Nikon TE2000 inverted microscope (Nikon, Melville, NY, USA). Microscopic images were recorded through a FITC filter from 1 min before to 7 min after drug treatment at a rate of two frames per sec. The intensity of intracellular green fluorescence of single cells was individually measured by the NIS-Elements AR 3.2 Software (Nikon, Melville, NY, USA).

## Immunocytochemistry

Cultured cells were fixed with PFA for 1 h and then washed with PBS. Cells were incubated for 1 day at 4˚C with a mouse monoclonal antibody against MAP2 (1:500) and then rinsed three times with PBS. The bound primary antibody was visualized using Alexa Fluor 488 goat anti-mouse secondary. Images were acquired using a monochrome camera Qi1-mc attached to a Nikon TE2000-E inverted microscope as previously described [14].

## In vitro Terminal Deoxynucleotidyl Transferase (TdT) -Mediated dNTP Nick End Labeling (TUNEL)

Cultures were assayed for DNA fragmentation using a TUNEL -based method as described by the manufacturer (In Situ Cell Death Detection Kit; Roche, Indianapolis, IN). Briefly, 4% PFA fixed cells were permeabilized in 0.1% Triton X-100 in 0.1% sodium citrate for 2 min on ice. To label damaged nuclei, 50 μL of the TUNEL reaction mixture was added to each sample and kept at 37˚C in a humidified chamber for 60 min. Controls consisted of not adding the label solution (terminal deoxynucleotidyl transferase) to the TUNEL reaction mixture.

The material was examined using a Nikon TE2000 inverted microscope equipped with fluorescence. TUNEL (+) cells were manually counted in 20× images (4 fields per well of 96-well plate).

## Animal surgery and drug administration

Rats were anesthetized with sodium pentobarbital (35 mg/kg, i.p.). A craniotomy of about 2–4 mm was made in the right squamosal bone. MCAo was induced by ligating the right distal MCA with a 10–0 suture using methods previously described [26]. After 60 min, the suture on the MCA and arterial clips on common carotids were removed to allow reperfusion. Core body temperature was monitored and maintained at 37˚C. After recovery from anesthesia, body temperature was maintained at 37˚C using a temperature-controlled incubator. Control animals received sham surgery, including craniotomy without MCAo. PM012 (50 mg) was added to saline (1 mL) and then vortexed for 1 min. PM012 (50 mg/kg/d x 2) or vehicle (saline) was given intraperinoneally from 2 days before the MCAo.

## Locomotor activity measurements

Animals were individually placed in $42 \times 42 \times 31$ cm open plexiglass boxes. Locomotor activity was recorded with an infra-red activity monitor (Accuscan, Columbus, OH) for 2 h (12-h light and 12-h dark/day) on day 2 (pretreatment experiment) after the MCAo [27]. The monitor contained eight vertical infrared sensors situated 10 cm from the floor of the chamber. Motor activities were calculated by the number of beams broken for 2 h after placement in the chamber. Vertical activity (VACTV; the total number of beam interruptions that occurred in the vertical sensors), total distance traveled (TOTDIST; the distance traveled in centimeters), and horizontal activity (HACTV; the total number of beam interruptions that occurred in the horizontal sensor) were analyzed by the Versamax program (Accuscan, Columbus, OH).

## 2,3,5-Triphenyltetrazolium Chloride (TTC) staining

Rats were decapitated 2 days after MCAo. The brains were removed and sliced into 2.0-mm sections. The brain slices were incubated in 2% TTC solution for 5 min at room temperature and then transferred into a 4% PFA solution for fixation. The area of infarction in each slice was measured with a digital scanner and the Image Tools program (University of Texas Health Sciences Center).

## Quantitative Reverse Transcription PCR (qRTPCR)

Cortical tissues from the infarcted and non-infarcted hemispheres were collected. Total RNAs were isolated using TRIzol Reagent (ThermoFisher, #15596–018, Waltham, MA, USA), and cDNAs were synthesized from 1 μg of total RNA by use of a RevertAid H Minus First-Strand cDNA Synthesis Kit (Thermo Scientific, #K1631, Waltham, MA, USA). cDNA levels for CD86, CD206, IBA1, IL-6, PERK, IRE1, CHOP, BIP, ATF6, caspase3, actin, and GAPDH were determined using specific universal probe library primer-probe sets or gene-specific primers (Table 1). Samples were mixed with TaqMan Fast Advanced Master Mix (Life Technologies, #4444557, Carlsbad, CA, USA) or SYBR (Luminaris Color HiGreen Low ROX qPCR Master Mix; ThermoScientific, Waltham, MA, USA). Quantitative realtime PCR (qRT-PCR) was carried out using the QuantStudio™ 3 Real-Time PCR System (Thermo Scientific,Waltham, MA, USA). The expression of the target genes was normalized relative to the endogenous reference genes (beta-actin and GAPDH averages) using a modified delta-delta-Ct algorithm. All experiments were carried out in duplicate.

**Table 1.  Oligonucleotide primers used for quantitative RT-PCR.**

| Gene | SYBR Green | | TagMan |
|---|---|---|---|
| | **Forward** | **Reverse** | |
| CD86 | TAGGGATAACCAGGCTCTAC | CGTGGGTGTCTTTTGCTGTA | |
| CD206 | AGTTGGGTTCTCCTGTAGCCCAA | ACTACTACCTGAGCCCACACCTGCT | |
| PERK | GAAGTGGCAAGAGGAGATGG | GAGTGGCCAGTCTGTGCTTT | |
| IRE1 | TCATCTGGCCTCTTCTCTCGGA | TTGAGTGAGTGGTTGGAGGC | |
| CHOP | ACCACCACACCTGAAAGCAG | AGCTGGACACTGTCTCAAAG | |
| Bip | TCGACTTGGGGACCACCTAT | GCCCTGATCGTTGGCTATGA | |
| ATF6 | GGACCAGGTGGTGTCAGAG | GACAGCTCTGCGCTTTGGG | |
| Caspase3 | GTGGAACTGACGATGATATGGC | CGCAAAGTGACTGGATGAACC | |
| IBA1 | | | Rn00574125_g1 |
| β-Actin | | | Rn00667869_m1 |
| GAPDH | | | Rn01775763_g1 |

### Statistics

Data are presented as the mean ± SEM. Unpaired t-test, one- or two-way ANOVA, and post-hoc Newman–Keuls (NK) test were used for statistical comparisons, with a significance level of $p < 0.05$.

## Acknowledgments

The authors thank Dr. Brandon Harvey at the National Institute on Drug Abuse for providing the AAV-GCaMP5 and Dr. Yun Wang at National Health Research Institutes for the critical comments.

## Author Contributions

**Conceptualization:** Seong-Jin Yu.

**Data curation:** Kuo-Jen Wu, Yu-Syuan Wang, Tsai-Wei Hung, Eun-Kyung Bae, Yun-Hsiang Chen.

**Investigation:** Seong-Jin Yu.

**Methodology:** Kuo-Jen Wu, Yu-Syuan Wang, Tsai-Wei Hung, Eun-Kyung Bae, Yun-Hsiang Chen, Dai-Won Yoo, Gyeong-Soon Kim.

**Resources:** Chan-Kyu Kim, Dai-Won Yoo, Gyeong-Soon Kim.

**Supervision:** Seong-Jin Yu.

**Visualization:** Tsai-Wei Hung, Eun-Kyung Bae.

**Writing – original draft:** Kuo-Jen Wu, Seong-Jin Yu.

**Writing – review & editing:** Seong-Jin Yu.

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
