## [Decision Letter · Decision Letter 0]

23 Nov 2022

PONE-D-22-30310Herbal formula PM012 induces neuroprotection in stroke brainPLOS ONE

Dear Dr. Yu,

Thank you for submitting your manuscript to PLOS ONE. After careful consideration, we feel that it has merit but does not fully meet PLOS ONE’s publication criteria as it currently stands. Therefore, we invite you to submit a revised version of the manuscript that addresses the points raised during the review process.

We look forward to receiving your revised manuscript.

Kind regards,

Cesar V Borlongan

Academic Editor

PLOS ONE

“This study was supported in part by Mediforum Co., Ltd., Seoul, Republic of Korea (C10-020).”

“C.K.K., D.W.Y., and K.S.K. are employees of Mediforum Co., Ltd. The National Health Research Institute and Mediforum Co., Ltd. have a Cooperative Research and Development Agreement to develop PM012 as a treatment strategy for neurodegenerative disorders. The authors declare that this study received partial funding from Mediforum Co., Ltd., Korea. The authors further declare that Mediforum Co., Ltd. provided input into dose selection and study design as well as editorial input into the article. The funder was not involved in the writing of this article, or the decision to submit it for publication.”

Please respond by return email with your amended Competing Interests Statement and we will change the online submission form on your behalf.

6. Please include your tables as part of your main manuscript and remove the individual files. Please note that supplementary tables (should remain/ be uploaded) as separate "supporting information" files

Additional Editor Comments:

Both reviewers are enthusiastic about this paper and raised only a few concerns/clarifications that can be addressed with minor revisions.

Reviewers' comments:

Reviewer's Responses to Questions

**Comments to the Author**

1. Is the manuscript technically sound, and do the data support the conclusions?

Reviewer #1: Yes

Reviewer #2: Yes

2. Has the statistical analysis been performed appropriately and rigorously? 

Reviewer #1: Yes

Reviewer #2: Yes

3. Have the authors made all data underlying the findings in their manuscript fully available?

Reviewer #1: Yes

Reviewer #2: Yes

4. Is the manuscript presented in an intelligible fashion and written in standard English?

Reviewer #1: Yes

Reviewer #2: Yes

5. Review Comments to the Author

Reviewer #1: The study by Dr Yu and colleagues show the potential of a herbal formulation to exert neuroprotection against experimental ischemic stroke. Using both in vitro and in vivo models of stroke, the authors demonstrate that PM12 protects against stroke-induced cell death with functional improvement, including reduction in brain damage and behavioral dysfunctions. These are straightforward results with careful and logical data interpretations and conclusions. Below are minor suggestions that may improve the study:

1. Providing insights into the active ingredients of PM12 that likely mediated the neuroprotective effects will likely provide interest.

2. For the cultured cell type and brain region of interest, the cortex was chosen. However, both cortical and non-cortical cells/tissues, such as the striatum, are damaged in ischemic stroke. Some insights into whether the authors also observed non-cortical protection by PM12 will be interesting. In the in vivo setting, did the authors observed protection of the striatum in addition to the cortex?

3. The experimental design employs a pre-treatment paradigm to show neuroprotection. For clinical application, a short discussion on how the authors will translate PM12 to at-risk stroke patients will be welcomed.

Reviewer #2: Dr Yu and co-authors utilized cell culture and animal models stroke to reveal the therapeutic effects of a herbal formula called PM12. The data show that PM12 exerts neuroprotection in cortical cells and in the cortex of stroke animals. Furthermore, cerebral infarction and locomotor impairments were reduced in PM-12-treated stroke rats. These findings suggest that therapeutic application of PM12 for neuroprotection against stroke. Overall, this is a very interesting study. I have a few simple questions that will only require the authors’ clarification.

1. The dose and timing of PM12 administration in the cell culture and animals will need some rationale. Were dose-response and time-dependent pilot studies conducted to arrive at these present doses and timing?

2. The intraperitoneal route of PM12 delivery in animals will also need clarification. What will be the envisioned clinical route? Also, a brief introduction on whether PM12 is BBB permeable will need to be included.

3. Figure 2, panel J will need lines between bars to indicate which groups are being compared with the “asterisks”.

4. Figure 3 will also need the addition of “asterisks” between the two groups to indicate significance in Ca++ fluoresecence.

5. For Figure 6, maybe instead of “lesioned” and “non-lesioned”, it is best to use “infarcted” and “non-infarcted” to refer to stroke brain pathology.

6. PLOS authors have the option to publish the peer review history of their article (what does this mean?). If published, this will include your full peer review and any attached files.

Reviewer #1: No

Reviewer #2: No

---

## [Author Response · Author response to Decision Letter 0]

13 Jan 2023

PONE-D-22-30310

Herbal formula PM012 induces neuroprotection in stroke brain

Dear Editor

We would like to thank the reviewers for the valuable comments. We also like to thank the editor to consider that our paper has merit for publication pending on the revision. We have addressed all the questions raised by the reviewers. Our responses and revision on the manuscript are summarized (in blue color) as follows. We hope that our revised manuscript will be fully acceptable by the journal.

Sincerely yours,

Seong-Jin Yu PhD

Editors' Comments

Reviewer #1: The study by Dr Yu and colleagues show the potential of a herbal formulation to exert neuroprotection against experimental ischemic stroke. Using both in vitro and in vivo models of stroke, the authors demonstrate that PM12 protects against stroke-induced cell death with functional improvement, including reduction in brain damage and behavioral dysfunctions. These are straightforward results with careful and logical data interpretations and conclusions. Below are minor suggestions that may improve the study:

A: We would like to thank reviewer 1 for considering our results are straightforward with careful and logical data interpretations.

1. Providing insights into the active ingredients of PM12 that likely mediated the neuroprotective effects will likely provide interest.

A: The crude ingredients of PM12 were included in paragraph 2 of page 3. Using an HPLC analysis, we found that PM12 contains bioactive molecules paeoniflorin and 5-hydroxymethylfurfural (Fig 1), which have potent anti-inflammatory (1) and anti-ER stress effects (2, 3). Inflammation and ER stress are major causes of neurodegeneration after ischemic brain injury. We and others reported that suppression of inflammation or ER stress improved neural functions in stroke animals (4, 5). These data suggest that paeoniflorin and 5-hydroxymethylfurfural are the active ingredients in PM012, which may induce protection against ischemic brain injury through anti-inflammation and anti-ER stress. These discussions were now included in the revised manuscript (page 9, paragraph 1) 

2. For the cultured cell type and brain region of interest, the cortex was chosen. However, both cortical and non-cortical cells/tissues, such as the striatum, are damaged in ischemic stroke. Some insights into whether the authors also observed non-cortical protection by PM12 will be interesting. In the in vivo setting, did the authors observed protection of the striatum in addition to the cortex?

A: In this study, a distal MCAo ligation model (i.e., occlusion of the right middle cerebral artery at the distal branch) was used to generate stroke in rats. The major lesion occurred in the ipsilateral cortex, as seen in the TTC staining (Fig 5). We did not examine the biochemical changes in the infarcted side striatum. However, in our recent unpublished observation, we found that PM012 induced protection in striatum in a 6-OHDA rodent model of Parkinson’s disease. The detailed protective mechanisms in striatum warrant further investigation. 

3. The experimental design employs a pre-treatment paradigm to show neuroprotection. For clinical application, a short discussion on how the authors will translate PM12 to at-risk stroke patients will be welcomed.

A: We demonstrated that pretreatment with PM012 reduced ischemic brain damage in rats. For clinical implication, this approach (pretreatment with PM012) may be beneficial to patients with a high risk of stroke (i.e., transient ischemic attack) to prevent recurrent ischemic injury. In our future experiments, we will examine the protective effect of PM012 when given early after stroke. (Please see page 10, paragraph 1).

Reviewer #2: Dr Yu and co-authors utilized cell culture and animal models stroke to reveal the therapeutic effects of a herbal formula called PM12. The data show that PM12 exerts neuroprotection in cortical cells and in the cortex of stroke animals. Furthermore, cerebral infarction and locomotor impairments were reduced in PM-12-treated stroke rats. These findings suggest that therapeutic application of PM12 for neuroprotection against stroke. Overall, this is a very interesting study. I have a few simple questions that will only require the authors’ clarification.

A: We would like to thank reviewer 2 for considering our study interesting.

1. The dose and timing of PM12 administration in the cell culture and animals will need some rationale. Were dose-response and time-dependent pilot studies conducted to arrive at these present doses and timing?

A: The doses of in vivo and in vitro experiments were determined by a dose-response study. Multiple doses of PM012 (0.1mg, 0.5mg, and 1mg) were used in cell culture (please see Fig. 1A). In addition, 26 rats were used to determine the dose of PM012 in vivo (veh, n=7; PM50mg, n=7, PM100mg, n=8, PM200mg, n=4). Our data demonstrated 50mg of PM012 was the most effective against ischemic stroke injury (Fig. 1B). 

The time of delivery was also evaluated in a pilot study. We found that PM012 was given 2 days before the MCAo induced the best protection. In our future study, we will examine the effectiveness and bioavailability of PM012 (see responses to Q2).

Fig. 1. Neuroprotective effects of the dose-response manner of PM012 in vitro (A) and in vivo (B).

2. The intraperitoneal route of PM12 delivery in animals will also need clarification. What will be the envisioned clinical route? Also, a brief introduction on whether PM12 is BBB permeable will need to be included.

A: PM012 was administered intraperitoneally to the experimental animals. In our future study, we will examine the effectiveness and bioavailability of PM012 after oral administration, as the oral route is more clinically relevant. 

The BBB is compromised shortly after the ischemic brain injury. It is possible that the active ingredients of PM012 can pass through the BBB in stroke brain.

3. Figure 2, panel J will need lines between bars to indicate which groups are being compared with the “asterisks”.

A: We have revised figure 2 accordingly.

4. Figure 3 will also need the addition of “asterisks” between the two groups to indicate significance in Ca++ fluoresecence.

A: We have revised figure 3 accordingly.

5. For Figure 6, maybe instead of “lesioned” and “non-lesioned”, it is best to use “infarcted” and “non-infarcted” to refer to stroke brain pathology.

A: We have revised figures 6 and 7 accordingly. Moreover, we have fixed it throughout the manuscript.

References

1. Zhang H, Jiang Z, Shen C, Zou H, Zhang Z, Wang K, et al. 5-Hydroxymethylfurfural Alleviates Inflammatory Lung Injury by Inhibiting Endoplasmic Reticulum Stress and NLRP3 Inflammasome Activation. Front Cell Dev Biol. 2021;9:782427.

2. Zhu Y, Han S, Li X, Gao Y, Zhu J, Yang X, et al. Paeoniflorin Effect of Schwann Cell-Derived Exosomes Ameliorates Dorsal Root Ganglion Neurons Apoptosis through IRE1alpha Pathway. Evid Based Complement Alternat Med. 2021;2021:6079305.

3. Zhu X, Wang K, Zhou F, Zhu L. Paeoniflorin attenuates atRAL-induced oxidative stress, mitochondrial dysfunction and endoplasmic reticulum stress in retinal pigment epithelial cells via triggering Ca(2+)/CaMKII-dependent activation of AMPK. Arch Pharm Res. 2018;41(10):1009-18.

4. Dong Z, Zhou J, Zhang Y, Chen Y, Yang Z, Huang G, et al. Astragaloside-IV Alleviates Heat-Induced Inflammation by Inhibiting Endoplasmic Reticulum Stress and Autophagy. Cell Physiol Biochem. 2017;42(2):824-37.

5. Hung TW, Wu KJ, Wang YS, Bae EK, Song Y, Yoon J, et al. Human Milk Oligosaccharide 2'-Fucosyllactose Induces Neuroprotection from Intracerebral Hemorrhage Stroke. Int J Mol Sci. 2021;22(18).

---

## [Editor Report · Decision Letter 1]

24 Jan 2023

Herbal formula PM012 induces neuroprotection in stroke brain

PONE-D-22-30310R1

Dear Dr. Yu,

We’re pleased to inform you that your manuscript has been judged scientifically suitable for publication and will be formally accepted for publication once it meets all outstanding technical requirements.

Kind regards,

Cesar V Borlongan

Academic Editor

PLOS ONE
---

## [Editor Report · Acceptance letter]

7 Feb 2023

PONE-D-22-30310R1 

Herbal formula PM012 induces neuroprotection in stroke brain 

Dear Dr. Yu:

I'm pleased to inform you that your manuscript has been deemed suitable for publication in PLOS ONE. Congratulations! Your manuscript is now with our production department. 

Kind regards, 

on behalf of

Prof. Cesar V Borlongan 

Academic Editor

PLOS ONE